# A Case History in Cooperative Biological Research: Compendium of Studies and Program Analyses in Kazakhstan

**DOI:** 10.3390/tropicalmed4040136

**Published:** 2019-11-09

**Authors:** Kenneth B. Yeh, Falgunee K. Parekh, Lyazzat Musralina, Ablay Sansyzbai, Kairat Tabynov, Zhanna Shapieva, Allen L. Richards, John Hay

**Affiliations:** 1MRIGlobal, Gaithersburg, MD 20878, USA; 2EpiPointe, LLC, Cary, NC 27518 USA; fparekh@epipointe.com; 3Department of Molecular Biology and Genetics, Al-Farabi Kazakh National University, Almaty 050040, Kazakhstan; Musralina@shh.mpg.de; 4Department of Biological Safety, Kazakh National Agrarian University, Almaty 050010, Kazakhstan; sansyzbai-ar@mail.ru; 5Department of Biological Safety, Kazakh National Agrarian University, Almaty 050010, Kazakhstan; kairat.tabynov@kaznau.kz; 6Department of Parasitic Diseases, Scientific Practical Center for Sanitary Epidemiological Expertise and Monitoring, Almaty 050008, Kazakhstan; 7Department of Preventative Medicine and Biostatistics, Uniformed Services University of the Health Sciences, Bethesda, MD 20814, USA; allen.richards@comcast.net; 8Department of Microbiology and Immunology, Jacobs School of Medicine and Biomedical Sciences, Buffalo, NY 14203, USA; jhay@buffalo.edu

**Keywords:** cooperative biological research (CBR) program, Kazakhstan, zoonoses, global health security, biosecurity, one health

## Abstract

Kazakhstan and the United States have partnered since 2003 to counter the proliferation of weapons of mass destruction. The US Department of Defense (US DoD) has funded threat reduction programs to eliminate biological weapons, secure material in repositories that could be targeted for theft, and enhance surveillance systems to monitor infectious disease outbreaks that would affect national security. The cooperative biological research (CBR) program of the US DoD’s Biological Threat Reduction Program has provided financing, mentorship, infrastructure, and biologic research support to Kazakhstani scientists and research institutes since 2005. The objective of this paper is to provide a historical perspective for the CBR involvement in Kazakhstan, including project chronology, successes and challenges to allow lessons learned to be applied to future CBR endeavors. A project compendium from open source data and interviews with partner country Kazakhstani participants, project collaborators, and stakeholders was developed utilizing studies from 2004 to the present. An earlier project map was used as a basis to determine project linkages and continuations during the evolution of the CBR program. It was determined that consistent and effective networking increases the chances to collaborate especially for competitive funding opportunities. Overall, the CBR program has increased scientific capabilities in Kazakhstan while reducing their risk of biological threats. However, there is still need for increased scientific transparency and an overall strategy to develop a capability-based model to better enhance and sustain future research. Finally, we offer a living perspective that can be applied to further link related studies especially those related to One Health and zoonoses and the assessment of similar capability-building programs.

## 1. Background and Introduction

After the Soviet Union breakup, the Nunn-Lugar Cooperative Threat Reduction program began in 1991 which funded work through the United States Department of Defense (US DoD) in several former Soviet Union (FSU) republics, including Kazakhstan, to dismantle, secure, and prevent proliferation of nuclear, chemical, and biological weapons of mass destruction (WMD). Since the early 2000s, the US DoD has funded and implemented a Biological Threat Reduction Program (BTRP) through the Defense Threat Reduction Agency (DTRA) in Kazakhstan.

After the BTRP addressed the biosecurity risks resulting from the breakup of the Soviet Union that included accounting for biological WMD and re-training scientists, the BTRP continued to fund the construction of a central reference laboratory (CRL), upgrades of other laboratories, and biological research. BTRP, which now runs programs in over 40 countries, has three lines of work: biosafety and biosecurity (BS&S); biosurveillance (BSV); and cooperative biological research (CBR). As discussed in our earlier manuscripts [1,2], Kazakhstan and the Central Asia region are interesting “hotspots” for infectious disease occurrence and surveillance due in part to its history, geography, and its diversity of vertebrate and invertebrate host species. Many of the BTRP supported research and epidemiologic studies involved One Health approaches, especially those studies related to zoonotic diseases. The objective of this project was to develop a historical perspective for the CBR program in Kazakhstan, including project chronology, successes and challenges, so that lessons learned might be applied to other CBR endeavors.

## 2. Historical Perspective

Kazakhstan has long maintained an infrastructure and tiered network for infectious disease surveillance that was a product of the Russian anti-plague (AP) system designed to combat seasonal epidemics through a multi-sectoral approach that recognized its zoonotic nature since the time of the Tsars [3,4]. The AP system consisted of a central administrative point which oversaw regional AP stations that, in turn, oversaw local field stations. The Russians recognized the need for “sentries” which served in the network to respond and mitigate outbreaks by deploying teams to the field by horse, covered wagon, and rail [4]. Dr. Aleksandr Gradzhanov, who was the director of the Uralsk Anti-Plague Station in western Kazakhstan, suggested these deployments were the first examples of mobile laboratories [5]. Later, some of the institutes from the AP system in Kazakhstan, which included the lead Almaty AP station, supported the Soviet bioweapons program [6]. Many of the foundational AP system organization, infrastructure, and resources still exist today at these institutes.

## 3. Economic Perspective

The infrastructure from the AP system also shaped the research culture among Kazakhstan and its neighbors in Central Asia which presented limited work often only in Russian language journals which explains why they are underrepresented in current scientific literature [7]. After the dissolution of the Soviet Union, Kazakhstan continued the AP system, but lost much of its funding and mission, resulting in an institutional “brain drain,” decay of infrastructure, and the biosecurity risk of unsecure pathogen material. Since Kazakhstan has become a republic, it has built a robust economy with its wealth of substantial natural resources: natural gas, minerals, and oil. In that regard, the International Monetary Fund, World Bank, and United Nations rank Kazakhstan in the middle fifty countries according to gross national product.

Research outputs are often measured in bibliometrics such as published and cited peer-reviewed articles. Kazakhstan maintains a high literacy rate among the general population, which reflects their historic scientific record of publications. The 2018 Nature Index ranks the top 50 countries for life sciences according to their research outputs and Kazakhstan is not ranked in the Nature Index, but it is ranked in the second tier of 50 countries at #76 according to the Scimago Institution under agricultural and biological sciences [8,9]. National public funding amounts and the number of publications are easily tracked and the public funding also influences the policies that shape the scope and definition of research [10]. Though the number of publications did increase in Kazakhstan during the period from 1996–2014, it can be argued that Kazakhstan’s current national policies for funding scientific research has not caught up with the current gross national product [7]. Thus, challenges remain among research programs in Kazakhstan including limited funding, improvement of infrastructure, need for development of programs that focus on merit-based advancement, fostering peer-reviewed quality research, and enhancing scientific transparency.

## 4. Overview of the CBR Program in Kazakhstan

Since 2005, DTRA BTRP has funded over $20 M in CBR alone in Kazakhstan mainly through single, large integrating contracts where contractors have implemented over 25 studies through their science partners (Figure 1). These studies were initially framed by the DTRA science leads, ministerial heads of Kazakhstan and subsequently the contractors developed a country science plan reflective of these leaders’ needs, which detailed the projects and studies, participating institutes, research collaborators, and stakeholders. Often the country plan described what studies would be implemented rather than describing an overall research strategy. Through the implementation of the CBR projects and studies, project outputs included developing abstracts and presentation of results, documenting progress in routine reports, and authoring peer-reviewed publications. In addition, outputs related to optimizing scientific methods were used to inform and revise standard operating procedures for infectious disease surveillance.

While the primary BTRP objective is related to US national security by reducing global health security threats by working with partner countries, CBR enables those partner countries to focus on broader capacity-building for biosurveillance and enhancing BS&S [11]. These capabilities include priorities for biosurveillance and pathogen research based on the US Department of Health and Human Services (DHHS) and the US Department of Agriculture (USDA) Select Agents list plus those pathogens of pandemic concern that include emerging infectious diseases; and promote hypothesis-based research projects. Earlier BTRP work in Kazakhstan and other FSU republics was implemented in a top-down fashion, through ministries and related state institutes such as public and veterinary health, which have disease surveillance responsibilities similar to the AP system. The list of participating institutes in Kazakhstan has grown along with the number of projects and studies (Table 1) since the BTRP program started, which is a positive aspect of the program.

The objectives for the three lines of BTRP’s work BS&S, BSV, and CBR are often achieved in consensus for capacity-building which is traditionally associated with international development, but are inherently difficult to measure. A major accomplishment of DTRA BTRP funding has been building research networks to better implement the CBR program in Kazakhstan, which can be measured in simple metrics such as conference presentations and peer-reviewed publications, both in the English language, and outcomes such as results from testing suspect samples and operational enhancements. The cooperation between Kazakhstan and the US continues to show intent for reducing biological threats and the CBR program has also increased cooperation among institutes in Kazakhstan that often did not previously communicate their respective infectious disease surveillance results. The overall capacity-building and capability-maturing has been harder to assess and quantify.

DTRA BTRP funded several CBR projects and smaller studies which contractors implemented during 2005–2007, 2009–2014, and 2015–2018. In 2005–2007, the initial CBR projects centered on studies involving select agents including zoonoses: anthrax, plague, tularemia, highly pathogenic avian influenza, and brucellosis. The typical CBR projects funded researchers in Kazakhstan and project collaborators in the US and UK who mentored and guided those researchers to develop and test their hypotheses. From 2009–2014, five CBR projects were funded with up to $ 1M each, which covered labor hours, materials, travel, and infrastructure overhead. In parallel, several smaller 12-month studies were funded with approximately $100 k called Threat Agent Detection and Response (TADR) Activity Projects (TAP) and were implemented in parallel to the CBR projects. The TAP studies were intended to support TADR which was the earlier name for the biosurveillance line of work and only funded material and did not fund labor hours. From 2015–2018, the trend for smaller studies continued as TAP studies evolved to Central Reference Laboratory (CRL) Activity Projects (CAP), which were designed and implemented to support studies for the new CRL that DTRA funded and constructed in Almaty, Kazakhstan.

CBR has served as a foundation for incorporating standard operating procedures into working practice under multiple research studies, especially related to BS&S to ensure that the work is performed in a safe and secure environment while supporting tasks to enhance infectious disease surveillance for human and animal health. More importantly, growing a quality research program especially in terms of capability maturity, that is the state “as is” and “to be”, requires a strategic vision. The evolution and progression of the Kazakhstan CBR program are linked and illustrated in an iterative fashion (Figure 1). The projects were largely assigned by the need and individual discussions among project participants, collaborators, and implementers, as this program was not developed through a traditional, competitive grant-based system. In 2014, from the original funding for 12 CBR projects and studies, the work completed lead to over six follow-up studies (Figure 1). From 2015–2018, later CBR projects were also developed and funded without earlier direct project links. There was a common focus to address research projects and studies that addressed zoonotic diseases and employed a One Health approach.

For example, KZ-29 a two-year CBR project (2012–2013) titled, Epidemiology of CCHF, HFRS (Hantavirus) and Tick-Borne Viral and Rickettsial Diseases in Kazakhstan, and a one-year TAP-2 study (2012) titled, Species Identification of Tick Vectors Associated with Infectious Disease in Kazakhstan, demonstrated the utility of multi-institute cooperation among several project collaborators the success of which led to multiple other projects (Figure 1). KZ-29 which, originated from developments of KZ-4, followed on with three additional TAP studies (TAP-10, Tick-borne encephalitis virus, *Coxiella*, and *Brucella* in milk; TAP-11, A Study of Hantavirus Genotypes Circulating in the West Kazakhstan Region; and CAP-1, Flea-borne disease surveillance) and another CBR project (KZ-31, titled Effect of Rickettsia spp. upon fitness of *Yersinia pestis* in fleas that vector plague in the Republic of Kazakhstan.

The multiple institutes involved in KZ-29 included three Kazakhstani institutes: (1) Scientific and Practical Center of Sanitary and Epidemiological Expertise and Monitoring (SPCSEEM), (2) Kazakh Scientific Center of Quarantine and Zoonotic Diseases (KSCQZD), and (3) the Ural’sk Anti-Plague Station (UAPS); and five international collaborating institutions: (1) University of Buffalo, (2) Naval Medical Research Center (NMRC), (3) Public Health England (PHE), (4) University of Florida and (5) United States Army Medical Research Institute of Infectious Diseases (USAMRIID). All of these institutions provided various skill sets and techniques from viral and rickettsial disease research fields, as well as data analysis and mapping. The project was successful, in obtaining many tick and human samples made available by the Kazakhstani scientists. Moreover, during the study a number of Kazakhstani institute staff was trained in modern diagnostic and data management techniques. Infectious disease agents were detected in multiple different areas of the country, several for the first time at specific locations, and the results were written up for presentations at a number of international meetings [12,13,14,15]. The downside for the project was a lack of sequencing capability on site and collaborators were unable to transfer any genomic material (including PCR products) out of the country, where sequence data could be generated. This lack of sequence data meant that otherwise important and novel data on new and suspected infectious agents could not be determined and therefore the epidemiology of the rodent- and vector-borne infectious diseases caused by these agents could not be properly defined. There were also issues of access to laboratories and difficulty in sharing data, while minor issues have potential to delay project schedule. Lastly, the incomplete investigation of the infectious agents made it difficult to obtain meaningful data and publish the results in peer-reviewed international journals. Nevertheless, the project revealed many potential infectious disease issues in Kazakhstan that will provide a starting point for future, in-depth investigations [2,16]. One of the solutions to these problems will be introduction on a national scale of identification workflows centered around rapid state-of-the-art genomic sequencing capability.

Additionally, some of the smaller TAP and CRL (CAP projects) studies demonstrated higher impact and success than the larger CBR projects because the project activity could be performed with a smaller number of researchers and required less coordination that would be necessary with researchers from various institutes in Kazakhstan. KZ-29 and TAP-2 were performed concurrently and leveraged the same project resources to fulfill both studies. The TAP-2 tick identification study which was designed to apply molecular methods, including using quantitative real-time polymerase chain reaction (qPCR) assays, to confirm the identification of ticks to the level of genus and species when the conventional visual identification was questionable due to inconclusive morphology and life stage presentation. Questing adult ticks were collected by tick drag during spring and summer from 2012–2014 in seven administrative rayons (county level) of West Kazakhstan oblast (province level). The total number of ticks collected (n = 2232) were identified according to visual morphological methods which resulted in four genera: *Dermacentor* (n = 2097), *Hyalomma* (n = 4), *Rhipicephalus* (n = 126), and *Ixodes* (n = 5). After DNA extraction, *Dermacentor* and *Ixodes* tick pools were tested using tick genus-specific assays that produced results in concordance with conventional visual identification [2]. Additional tests were run to exclude cross-reaction and ensure fidelity of the *Dermacentor* and *Ixodes* assays. A partner country scientist from UAPS presented this data in a poster at the 2013 Conference of Research Workers in Animal Diseases (CRWAD) and demonstrated the utility of a molecular PCR-based method to complement and enhance tick identification [12]. The KZ-29 and TAP-2 examples demonstrated synergy through linking the studies in a practical manner.

In terms of simple metrics, from 2009–2014, the CBR, TAP, and CAP work resulted in at least 12 peer-reviewed publications, 60 conference presentations, and 30 research collaborator visits to Kazakhstan. Overall, the Kazakhstan CBR projects have resulted in a few hundred conference presentations and several peer-reviewed publications. The benefit of conference presentations is the practice of developing and presenting results while meeting and networking with peers in the scientific community. While the number of peer-reviewed publications is often used to judge scientific merit, publications are often developed well after a project has ended. In addition, the number of citations from these publications also grew over time beyond the time of the initial publications (Table 2 and Table 3).

More recent CBR work has engaged applied science and academic institutes in Kazakhstan to take advantage of their capabilities and resources for next-generation workforce training and tapping into their basic research programs. These resources also include individuals who have been educated through the current system that includes more exposure to US and Western science and technology and better knowledge of the English language associated with major publications. This approach also reinforces similar examples of public–private partnerships where a “triad” is formed when academic and university sectors are engaged [17]. In Kazakhstan, these capabilities and resources complemented the CBR program while their peer institutes, already engaged in this program, have the required infrastructure to perform research activities on Select Agents which require higher biological containment measures. As the CBR program has evolved, the next-generation workforce has also developed through experience and incorporating various participants such as academics. It has become apparent to the US participants that levels of English language proficiency among Kazakhstani collaborators has markedly increased. This is an obvious factor in improving better communications and has a larger role in project planning and implementation. In the long term, this will have been one of the most useful by-products from BTRP, enabling Kazakhstani personnel access to international literature, scientific meetings, and personal relationships with international scientists and physicians. This ability and practice to consistently and effectively network increases the chances to collaborate especially for competitive funding opportunities.

## 5. Conclusions

Reducing biological threats through CBR programs is still important even in the absence of biological weapons programs. BTRP has specific objectives for reducing biological threats especially WMD and emerging infectious diseases as it relates to US national security and the research funded is a hybrid of pathogen research and disease surveillance [11]. The cooperation and investments in CBR have led to improved infrastructure, resources, and operations for enhancing active surveillance for infectious disease outbreaks, whether natural, intentional, or accidental. In our analysis, findings related to the CBR program in Kazakhstan include successes, limitations, and continuing challenges that are summarized from recent discussions and comparative literature reviews. The CBR program, which so far has demonstrated data-sharing in Kazakhstan can be an effective model for enhancing cooperation and capability-building which in turn reduces the risk for potential future biological threats.

Scientific accomplishments and capability-building were also demonstrated in Kazakhstan through the CBR program despite challenges and limitations. In addition to simple metrics, the nature of the respective collaborations is important to consider as it often influences future outputs such as when publications are made and how further work is proposed and implemented. In Kazakhstan, we also observed that when work is published the impact on citations is not known for some time and often the publication was not the primary reason for collaboration [18]. Limited feedback from Kazakhstani CBR participants also stated the opportunity to receive external funding for research and publishing this work were likely the primary reason for engaging in the CBR program.

At a 2018 National Academies of Sciences, Engineering, and Medicine workshop called Enhancing Global Health Security through International Biosecurity and Health Engagement Programs, a committee was formed to assist BTRP and provide future guidance. One element related to CBR and capability-building in general is the need to exact expectations, which is important under a highly specific program that may have different views across the funder, implementers, and partner country participants [19]. Exacting expectations across multiple stakeholders is especially difficult when the program requirements and policy do not result in a one-size-fits-all approach. An agile approach that reinforces existing resources and uses a multi-disciplinary and multi-sectoral approach is effective for capability and development.

Continuing challenges for increasing scientific transparency and incorporating a capability maturity model in order to advance respective research programs were also noted. In addition to the KZ-29 and TAP-2 examples described, the importance and difficulty for increasing scientific transparency via sample material exchange was noted. While the exchange of genomic sequences has been demonstrated, working towards sample material exchange would further increase scientific transparency [19]. BTRP and other programs have funded work related to genomic sequencing and infectious disease modeling [20,21]. These programs funded by the US and other countries worldwide further encourage cooperation and may present opportunities to exchange sample material.

For long-term capability-building, the sustainability of CBR program in Kazakhstan needs to reach a maturity level where it can compete and acquire external research funding apart from BTRP; this threshold remains a challenge. When funding a scientific research program, it is difficult to balance immediate needs and manage risk with raising quality and innovation. Similarly, standardized evaluation techniques such as the capability maturity model (CMM) which are often used in business and technology sectors are also good for assessment at a scientific research program level. At the BTRP level, outputs for cost-effectiveness, which could be incorporated into their program lifecycle analysis, and measuring benefits to foreign relations would augment program metrics. A unified top-down and bottom-up approach that is focused on capability maturity also needs to be employed to ensure that funder and recipients agree on expectations and science objectives.

The CBR program continues to align outputs for long-term sustainability such as encouraging partner countries to establish their own external funding. From our literature review, sustainability and setting policy are two common challenges similar programs also face [22]. For BTRP, designing impactful projects, which would reinforce their targeted scientific engagement approach, is even more challenging given specific program objectives and short project durations. It is suggested that the overall impact is limited when a linear process starting with the science integrity and ending with applying those findings and outcomes to a real world [22]. It is well known that multi-sectoral collaborations are effective and that motivations among collaborators affect the research outputs [18]. Within specific program objectives, certain requirements for research proposals should be viewed more as opportunities rather than limitations [19]. Going forward, the ability for Kazakhstani partner participants to further develop research proposals for funding, leverage existing resources and networks will continue to enhance the CBR program.
Publicly posted funding calls through major national government funders such as BTRP broad area announcements are prime examples of external funding that Kazakhstan can apply for and pursue. While BTRP’s fundamental research scope is on US Biological Select Agents, pathogens of pandemic potential, emerging, and re-emerging infectious diseases, partner country scientists and research collaborators have linked non-Select Agents to research topics through developing hypotheses for co-infection and differential diagnosis.Leveraging existing resources including knowledge of in-country organizational structure, workflow, related networks, and current studies are important to establish buy-in from a top administrative level to the working research level which further reinforces multi-sectoral approach and encourages communication.The degree of collaboration in Kazakhstan is emphasized especially in teamwork to develop strategies and cooperate effectively. Leveraging existing resources while reinforcing networks and maintaining relationships should be frequent and encouraged. The ability to link consecutive projects through effective collaboration and funding pursuits builds greater capability and avoids one-off studies.

Our experience suggests incorporating multi-sectoral partners that better leverage existing resources may offer higher impact especially among similar research studies of small scale and magnitude [19]. For example, initial studies such as those involving multi-institute participants in Kazakhstan were challenged with distributing project tasks and budgets equally. Often these initial studies were designed to demonstrate new cooperation among institutes in Kazakhstan, however they may have been performing similar tasks under their respective missions. Alternatively, a multi-sectoral approach such as those providing mutual benefits across different scientific, public health communities, and disadvantaged populations have benefits beyond generating research data [23]. An important effect of employing these types of studies is engaging the public and educating them on the purpose and objective of this work as related to larger programs which demonstrates transparency and counters any misperceptions and stigmas. Further incorporating approaches such as One Health that included joint animal and human health studies in Kazakhstan should also be encouraged to increase impact and reinforce cooperation in future CBR studies.

Our compendium provides a perspective that can help draw a roadmap for a scientific research program when the project outputs and metrics are incorporated. In addition, aligning program objectives and exacting expectations across participants and stakeholders contribute to the roadmap that ultimately builds that partner country’s capability and capacity necessary for sustainability. Tracking the evolution of program projects and studies, especially collaborations and networks, is emphasized to capture important activities that are often omitted when large programs evolve and institutional memory is lost through turnover of human resources. To advance science through increased transparency and mature capabilities, a multi-sectoral approach, commitment, and vision is required to advocate a strategy and can be implemented through strong leadership.

## Figures and Tables

**Figure 1 tropicalmed-04-00136-f001:**
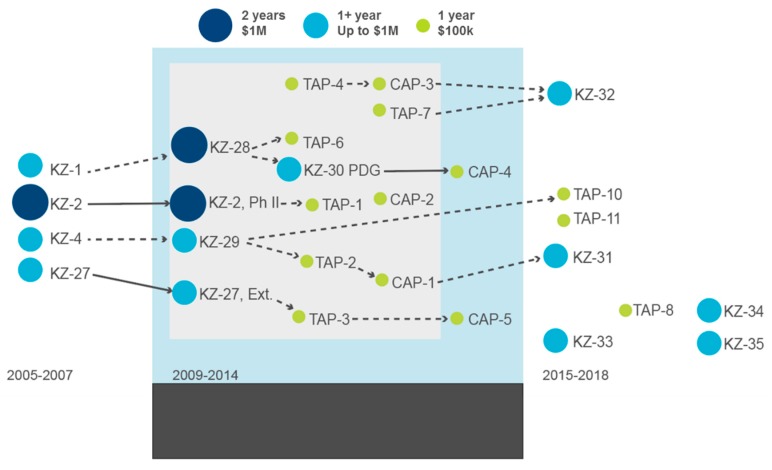
Project history map of the CBR program in Kazakhstan.

**Table 1 tropicalmed-04-00136-t001:** List of Kazakhstani institutes participating in cooperative biological research (CBR).

Institute Name	Ministry	Mission
Kazakh Scientific Center of Quarantine and Zoonotic Diseases (KSCQZD)	Ministry of Health (MOH)	Lead science research center for Kazakh anti-plague stations.
National Center for Biotechnology (NCB)	Ministry of Education and Science (MOES)	Stimulate science for agricultural, environmental, and health applications; biotechnology transfer.
National Reference Veterinary Center (NRVC)	Ministry of Agriculture (MOA)	Control and surveillance of infectious diseases that affect animals and livestock.
Research Institute for Biological Safety Problems (RIBSP)	Ministry of Education and Science (MOES)	Performs research on various pathogens of animals and plants.
Scientific and Practical Center of Sanitary and Epidemiological Expertise and Monitoring (SPCSEEM)	Ministry of Health (MOH)	Control and surveillance of communicable and non-communicable diseases that affect people; protection of environment from pathogens.
Uralsk Anti-Plague Station (UAPS)	Ministry of Health (MOH)	Regional center for control of human infectious diseases.

**Table 2 tropicalmed-04-00136-t002:** Summary of CBR studies: 11 projects (1–2 year, $0.5M–2M) and 15+ studies (1 year, $100k). Several studies included zoonoses, One Health approaches and geographic information systems, which were favorably received at the international meetings.

Topic	Project Number	Major Outcomes/Accomplishments
Multi-pathogen, zoonotic diseases	KZ-1 (anthrax); KZ-28 (anthrax, plague, and tularemia); TAP-6 (plague)	5 publications; KZ-28 collaborators conducted 2 visits to Kazakhstan; Kazakhstani scientists gave at least 2 presentations.
Brucellosis	KZ-2 (brucellosis); TAP-1 (qPCR methods); CAP-2 (milk)	1 publication; KZ-2: collaborators conducted 11 visits to KZ institutes; Kazakhstani scientists gave 19 presentations (10 conferences).
Viral and rickettsial vector-borne and zoonotic diseases	KZ-4, KZ-29 (Crimean-Congo Hemorrhagic Fever and Tick-Borne Encephalitis viruses, hantavirus, and rickettsial disease); TAP-2 (tick identification), CAP-1 (flea-borne disease); TAP-10 (TBE); KZ-31 (rickettsia)	2 publications; KZ-29 collaborators conducted 14 visits to Kazakhstan; collaborators also hosted sequencing workshop; Kazakhstani scientists gave 21 presentations (11 conferences).
Bird diseases	KZ-27, CAP-5 (avian influenza); TAP-3, TAP-12 (NDV)	KZ-27 collaborators conducted 3 visits to Kazakhstan, 1 collaborator hosted workshop; Kazakhstani gave 4 presentations (3 conferences).
Animal diseases	TAP-4 (DIVA foot and mouth disease virus); CAP-3 (Bluetongue, Akabane, Schmallenberg surveillance); TAP-7 (African Swine Fever education outreach); TAP-8 (Saiga mortality); KZ-30, CAP-4 (Capripox sequencing); KZ-32 (Bluetongue virus, Brucella); KZ-33 (Middle Eastern respiratory syndrome coronavirus); KZ-35 (highly pathogenic swine fever viruses)	Various collaborators conducted visits; Kazakhstani scientists gave at least one presentation for each project.

**Table 3 tropicalmed-04-00136-t003:** List of Biological Threat Reduction Program (BTRP) funded projects in Kazakhstan and publications. The publications listed reflects those authored by Kazakhstani, US, and UK collaborators published in English language journals including number of times cited [x] via Google search (1 June 2019).

Project	Topic	Publications and Selected Abstracts
KZ-1	anthrax	Mullins JC, Garofolo G, Van Ert M, Fasanella A, Lukhnova L, Hugh-Jones ME, Blackburn JK. Ecological niche modeling of *Bacillus anthracis* on three continents: evidence for genetic-ecological divergence? PloS one. 2013 Aug 19;8(8):e72451.e72451. https://doi.org/10.1371/journal.pone.0072451.
		Kracalik IT, Blackburn JK, Lukhnova L, Pazilov Y, Hugh-Jones ME, Aikimbayev A. Analysing the spatial patterns of livestock anthrax in Kazakhstan in relation to environmental factors: a comparison of local (Gi*) and morphology cluster statistics. Geospatial Health. 2012;7(1):111–26.
		Mullins J, Lukhnova L, Aikimbayev A, Pazilov Y, Van Ert M, Blackburn JK. Ecological Niche Modelling of the *Bacillus anthracis* A1.a sub-lineage in Kazakhstan. BMC ecology. 2011 Dec;11(1):32.
		Joyner TA, Lukhnova L, Pazilov Y, Temiralyeva G, Hugh-Jones ME, Aikimbayev A, Blackburn JK. Modeling the potential distribution of *Bacillus anthracis* under multiple climate change scenarios for Kazakhstan. PloS one. 2010 Mar 9;5(3):e9596.
		Aikembayev AM, Lukhnova L, Temiraliyeva G, Meka-Mechenko T, Pazylov Y, Zakaryan S, Denissov G, Easterday WR, Van Ert MN, Keim P, Francesconi SC. Historical distribution and molecular diversity of *Bacillus anthracis*, Kazakhstan. Emerging Infectious Diseases. 2010 May;16(5):789.
KZ-2	Brucellosis	Mizanbayeva S, Smits HL, Zhalilova K, Abdoel TH, Kozakov S, Ospanov KS, Elzer PH, Douglas JT. The evaluation of a user-friendly lateral flow assay for the serodiagnosis of human brucellosis in Kazakhstan. Diagnostic microbiology and infectious disease. 2009 Sep 1;65(1):14–20.
KZ-2, phase II	Brucellosis	Abstracts: Omasheva, G., Aikimbayev, A., Zhandossov, Sh., Tuleov, A., Hagius, S., Elzer, P., Nikolich, M.P., Blackburn, J.K. Brucellosis in Kazakhstan. 66th Annual Brucellosis Research Conference, 7–8 December 2013, Chicago, IL.
		Syzdykov, M.S., Kuznetsov, A.N., Sadovskaya, V.P., Blackburn, J.K. Spatial analysis of the brucellosis distribution in southeastern Kazakhstan using GIS technologies. URISA GIS in Public Health Conference, Miami, FL 17–20 June 2013.
		Sydyzkov, M.S., Kuznetsov, A.N., Huang, X., Elzer, P.H., Espembetov, B.A., Daulbayeva, S.F., Blackburn, J.K., Nikolich, M.P. Evaluation of spatial patterns of brucellosis in southern Kazakhstan using GIS technologies. 66th Annual Brucellosis Research Conference, 7–8 December 2013, Chicago, IL.
		Sytnik, I., Tyulegneov, S., Karibayev, T., Dzhailbekova, A., Shcherbakov, A., Seidakhmetova, R., Abenova, A., Nikolich, M., Elzer, P., Blackburn, J.K., Huang, X. Ecology of Brucella biotypes in southern Kazakhstan. 66th Annual Brucellosis Research Conference, 7–8 December 2013, Chicago, IL.
		J. Blackburn, M. Nikolich, P. Elzer, X. Huang, G. Omasheva, A. Aikimbayev. Results and Prospects of Brucellosis Research in Kazakhstan. Brucellosis Research Conference, Chicago, IL, December 1,2, 2012.
		Syzdykov, M.S., B.B. Atshabar, S.V. Kazakov, A.N. Kuznetsov, T.A. Grushina, S.F. Daulbaeva, J.K. Blackburn, S. Mizanbayeva. Development of a GIS-based surveillance system to monitor human brucellosis in Kazakhstan. Brucellosis 2011 International Research Conference, Puerto Madero, Buenos Aires, Argentina, September 21–23,2011.
		T. Grushina, P. Elzer, J.K. Blackburn, X.Huang, B. Atshabar, O. Karpova, M. Syzdykov, S. Daulbayeva, A. Kuznetsov, K. Ospanov, S. Kazakov, S. Mizanbayeva, D. Akzholtaeva, G. Mukhamadiyanova, G. Kalykova, T. Karibayev, S. Tyulegenov, D. Berezovskiy, M. Nikolich. Evaluation of Multiple-Locus Variable-Number Tandem-Repeat Analysis Method for Genotyping Human Brucella isolates in Kazakhstan. Brucellosis 2011 International Research Conference, Puerto Madero, Buenos Aires, Argentina, September 21–23, 2011.
		J. Blackburn, M. Nikolich, P. Elzer, X. Huang, G. Omasheva, A. Aikimbayev. Results and Prospects of Brucellosis Research in Kazakhstan. Brucellosis Research Conference, Chicago, IL, December 1,2, 2012.
		J.K. Blackburn, G. Kazakova, M.J. Wilson, T.A. Joyner, I.T. Kracalik, P. Elzer, G. Mukhamadinov, M.P. Nikolich, I. Sytnik. Exploring spatio-temporal shifts of human and livestock brucellosis using serological surveillance in Kazakhstan 2007—2008. Brucellosis Research Conference, Chicago, Illinois 4–6 December 2010.
KZ-28	Multi-pathogen, zoonotic diseases: anthrax, plague, and tularemia	Abstract: Sadovskaya, V.P., Atshabar, B.B., Kazakov, S.V., Burdelov, L.A., Zhumadilova, Z.B., Syzdykov, M.S., Blackburn, J.K. Employing GIS and spatial analysis to inform plague surveillance in Kazakhstan. URISA GIS in Public Health Conference, Miami, FL 17–20 June 2013.
KZ-29	Vector-borne viral and rickettsial disease	Nurmakhanov T, Sansyzbaev Y, Atshabar B, Deryabin P, Kazakov S, Zholshorinov A, Matzhanova A, Sadvakassova A, Saylaubekuly R, Kyraubaev K, Hay J. Crimean-Congo haemorrhagic fever virus in Kazakhstan (1948–2013). International Journal of Infectious Diseases. 2015 Sep 1;38:19–23.
		Hay J, Yeh KB, Dasgupta D, Shapieva Z, Omasheva G, Deryabin P, Nurmakhanov T, Ayazbayev T, Andryushchenko A, Zhunushov A, Hewson R, Farris CM and Richards AL (2016) Biosurveillance in Central Asia: Successes and Challenges of Tick-Borne Disease Research in Kazakhstan and Kyrgyzstan. *Front. Public Health* 4:4. doi:10.3389/fpubh.2016.00004.
		Abstracts: Andryushchenko AV, Ayazbayev TZ, Bidashko FG, Tanitovsky VA, Farris CM, Richards AL. Detection of rickettsial DNA from Ixodid ticks of the West Kazakhstan region. ASM 2014 Boston, MA. 17–20 May 2014. Abst: #852.
		Kyraubayev K, Shapiyeva Z, Utegenova U, Zhandosov S, Beysenaeva M, Ziyadina L, Omasheva G, Hay J, Farris CM, Richards AL. Study of *Dermacentor marginatus* ticks for rickettsiae in Central Kazakhstan. ASM 2014 Boston, MA. 17–20 May 2014. Abst: #858.
		Nurmakhanov T, Sansyzbayev Y, Yeskhodzhayev O, Vilkova A, Berdibekov A, Matzhanova A, Sailaubek R, St John H, Farris CM, Richards AL. Presence of tick-borne *Rickettsia* pathogens in southern Kazakhstan. ASM Microbe. Boston June 16–20, 2016.
TAP-2	Species identification of tick vectors	Andryushchenko A, Ayazbayev T, Richards A, Pisarcik S. Tick identification in Northwestern Kazakhstan using morphological and molecular characteristics. International Journal of Infectious Diseases. 2014 Apr 1;21S:393. Abstract ICID
KZ-31	Rickettsia and plague in fleas	Sansyzbayev Y, Nurmakhanov T, Berdibekov A, Vilkova A, Yeskhodzhayev O, St. John HK, Jiang J, Farris CM, Richards AL. Survey for Rickettsiae Within Fleas of Great Gerbils, Almaty Oblast, Kazakhstan. Vector-Borne and Zoonotic Diseases. 2017 Mar 1;17(3):172–8.
		Nurmakhanov T, Sansyzbayev Y, John HS, Farris C, Richards A. Flea-Borne Rickettsiae in Almaty Oblast, Kazakhstan. Online Journal of Public Health Informatics. 2016 Mar 24;8(1). Abst.
		Yerubayev T, Nurmakhanov T, Meka-Mechenko T, Abdirassilova A, Yeskhojayev O, Vilkova A, Ussenbekova D, Richards A, Farris C, Motin V. Investigating the presence of *Rickettsia* spp. and *Yersinia pestis* in flea from the natural plague foci of Kazakhstan. 30th Meeting of the American Society for Rickettsiology, Santa Fe, NM June 8–11, 2019. Poster. Abst #15. KZ-31
KZ-33	MERS-CoV surveillance	Mendenhall IH, Kerimbayev AA, Strochkov VM, Sultankulova KT, Kopeyev SK, Su YCF, Smith GJD, Orynbayev MB. Discovery and Characterization of Novel Bat Coronavirus Lineages from Kazakhstan. Viruses. 2019 Apr 17;11(4).
TAP-10	Tick-borne encephalitis virus, Coxiella, and Brucella in milk	Hay J, Farris C, Elzer P, Andrushchenko A, Hagius S, Richards A, Ayazbayev T. Tick-Borne Encephalitis Virus, *Coxiella burnetii* & *Brucella* spp. in Milk, Kazakhstan. Online Journal of Public Health Informatics. 2016 Mar 24;8(1). Abst.

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
