# Peer review of "A Case History in Cooperative Biological Research: Compendium of Studies and Program Analyses in Kazakhstan"

_tropicalmed, 2019, doi:10.3390/tropicalmed4040136_

Round 1
Reviewer 1 Report
A case history in cooperative biological research: compendium of studies and program analyses in Kazakhstan
Thank you for the opportunity to review this paper. This manuscript provides a comprehensive review of the cooperative biological research program in Kazakhstan and attempts to capture the impact of many years of investment by the U.S. Department of Defense. This is a contribution to the literature, in that few papers exist which provide a complete overview of a program. There are a few issues, however, that would be helpful for the authors to address:
The authors discuss the metrics for success for the research program, measured mostly by outputs and collaborations. Would be helpful to discuss other potential metrics of success, even if the data are not available to assess. Such as cost benefit/cost effectiveness analysis. How would you measure that? A lot of money has gone into this program…is it worth it? Can we document the cost effectiveness? What about diplomacy/ foreign relations? How has the program and collaborations contributed in that realm? Is that measurable? If not, should it be? The paper emphasizes the importance of English language publications and language proficiency. It might be helpful to add additional language on why that is important in the scientific community, beyond just being easier for US and UK collaborators. The authors allude to the international importance, but still worth a bit more description Is there a clearer way to present figure 1? The information is there, but might be cleaner ways to visualize the data Authors discuss ‘minor issues of access to laboratories and difficulty in sharing data’. (page 6 and then revisited in the discussion) I would suggest these are not minor issues and maybe CBR can be a model for how to develop strong access and benefit sharing regimes/MTAs that respect all parties. A comment- I noted the publications tend to have first authors that are from the US or UK. Can the authors comment on what that indicates for collaborations and development of true partnerships/capacity? Is there a trend where authorship shifts over time? Who is first and last can indicate a lot about a research collaboration. Even on this very paper. When should a program ‘graduate’? What does sustainability really look like?
Author Response
Author’s reply to Reviewer 1:
The authors discuss the metrics for success for the research program, measured mostly by outputs and collaborations. Would be helpful to discuss other potential metrics of success, even if the data are not available to assess. Such as cost benefit/cost effectiveness analysis. How would you measure that? A lot of money has gone into this program…is it worth it? Can we document the cost effectiveness? What about diplomacy/ foreign relations? How has the program and collaborations contributed in that realm? Is that measurable? If not, should it be?
Edited line 121-122 to qualify building research networks and edited lines 318-320: The ability to link consecutive projects through effective collaboration and funding pursuits builds greater capability and avoids one-off studies.
Edited lines 134-136 to better describe what CBR project funds.
Edited line 285-286: At the BTRP perspective, outputs for cost-effectiveness, which could be incorporated into their program lifecycle analysis, and benefits to foreign relations benefits would augment program metrics.
Note the DOD tracks program lifecycle by country and doing cost-benefit and effectiveness analysis would be their discretion. Otherwise, our work demonstrates the effectiveness of networks and collaborations to build research through consecutive projects rather than one-off studies.
Edited lines 286-288, and 294 to address cost benefit and measure benefits to foreign relations.
The paper emphasizes the importance of English language publications and language proficiency. It might be helpful to add additional language on why that is important in the scientific community, beyond just being easier for US and UK collaborators.
Comment acknowledged and text edited (line 123) to reinforce importance of English language in science and medicine (mentioned in Table 3 caption and lines 231, 238.
The authors allude to the international importance, but still worth a bit more description Is there a clearer way to present figure 1? The information is there, but might be cleaner ways to visualize the data
Since various contractors typically implement the project studies, the challenge has been capturing past work and linking it with current work. We believed the bubble chart was the most effective way to show the project linkages rather than focus on project topics and collaborators. Comment is acknowledged and we will consider alternate ways to illustrate the data in the future iterations.
Authors discuss ‘minor issues of access to laboratories and difficulty in sharing data’. (page 6 and then revisited in the discussion) I would suggest these are not minor issues and maybe CBR can be a model for how to develop strong access and benefit sharing regimes/MTAs that respect all parties.
Appreciate suggesting the link that CBR is a model for sharing. Edited lines 186-187 for access issues and 254-255 for data sharing.
A comment- I noted the publications tend to have first authors that are from the US or UK. Can the authors comment on what that indicates for collaborations and development of true partnerships/capacity? Is there a trend where authorship shifts over time? Who is first and last can indicate a lot about a research collaboration. Even on this very paper.
Excellent point on true collaborations etc. (thanks) which we’ll address in the next iteration. The authors have the authorship trend in mind and we believe it will shift to primarily Kazakhstani authorship in the future as part of our coaching and mentorship. The “showing” then “doing” training approach has been shown in our work and evident in this a paper. We will address this trend and authorship evolution in our next manuscript.
When should a program ‘graduate’? What does sustainability really look like?
Another excellent point related to the capability maturity model and we suggest that the ability to secure external funding is a measurement of graduating from the BTRP program. The authors plan to expand on this in the next manuscript.
Reviewer 2 Report
A very useful publication, particularly the thoughts of these experienced 'engagers' on common-sense 'metrics'. Metrics have been a challenge for the CTR program over the years; they have often tried to make them too detailed and comprehensive. It's possible to succeed in implementing such metrics and still fail in the engagement. Also appreciate some of the observations on the challenges and value of building relationships between Kazakhstani scientist and scientists from US and UK. Finally, the comment about the empowering value of just improving English capabilities and understanding of western 'ways' in science is an excellent observation. Many of these activities described are often overlooked, but are much more valuable than simply 'consolidating pathogens' and putting a tall fence around labs working with 'select agents'. The human networks and relationships established through these programs are extremely valuable outcomes.
Author Response
Thank you and we appreciate your insight and supportive feedback.
Reviewer 3 Report
Dear Authors
The purpose of a review paper is to succinctly review recent progress in a particular topic. Overall, the paper provides a historical perspective for the CBR involvement in Kazakhstan, including project chronology, successes and challenges, in an One Health perspetive.
The study is correctly designed and technically sound. The work provide an advance towards the current knowledge. The revision from a scientific point of view seems to be well done.
References cited are recent and have a high relevance to the problem.
Author Response

(The authors gave the same response as above.)
